# Scoping Review of Intervention Strategies for Improving Coverage and Uptake of Maternal Nutrition Services in Southeast Asia

**DOI:** 10.3390/ijerph182413292

**Published:** 2021-12-16

**Authors:** Kauma Kurian, Theophilus Lakiang, Rajesh Kumar Sinha, Nishtha Kathuria, Priya Krishnan, Devika Mehra, Sunil Mehra, Shantanu Sharma

**Affiliations:** 1MAMTA Health Institute for Mother and Child, Delhi 110048, India; kurian.kauma@gmail.com (K.K.); theo.lakiang@gmail.com (T.L.); cmarajesh@gmail.com (R.K.S.); nishthak@mamtahimc.in (N.K.); priyakrishnan76@gmail.com (P.K.); devika.mehra@mamtahimc.in (D.M.); dr_mehra@mamtahimc.in (S.M.); 2Department of Clinical Sciences, Lund University, Skåne University Hospital, S-20502 Malmo, Sweden

**Keywords:** community mobilization, counselling, coverage, home visits, maternal nutrition

## Abstract

Maternal undernutrition can lead to protein-energy malnutrition, micronutrient deficiencies, or anemia during pregnancy or after birth. It remains a major problem, despite evidence-based maternal-nutrition interventions happening on ground. We conducted a scoping review to understand different strategies and delivery mechanisms to improve maternal nutrition, as well as how interventions have improved coverage and uptake of services. An electronic search was conducted in PubMed and Google Scholar for published studies reporting on the effectiveness of maternal-nutrition interventions in terms of access or coverage, health outcomes, compliance, and barriers to intervention utilization. The search was limited to studies published within ten years before the initial search date, 8 November 2019; later, it was updated to 17 February 2021. Of 31 studies identified following screening and data extraction, 22 studies were included for narrative synthesis. Twelve studies were reported from India and eleven from Bangladesh, three from Nepal, two from both Pakistan and Thailand (Myanmar), and one from Indonesia. Nutrition education and counselling, home visits, directly observed supplement intake, community mobilization, food, and conditional cash transfer by community health workers were found to be effective. There is a need to incorporate diverse strategies, including various health education approaches, supplementation, as well as strengthening of community participation and the response of the health system in order to achieve impactful maternal nutrition programs.

## 1. Introduction

Maternal nutrition refers to increased nutritional demand during adolescence, antenatal, and postnatal periods. A shortfall in nutrition during these periods can lead to protein-energy malnutrition, micronutrient deficiencies, poor weight gain, or anemia [1]. In low- and middle-income countries, 450 million women are estimated to have short stature, 240 million are underweight (Body Mass Index-BMI < 18.5 kg/m^2^), and 468 million are anemic [2]. This could be attributed, in part, to the poor performance of maternal nutrition programs in low- and middle-income countries. For example, only 31% of pregnant women in low- and middle-income countries consumed iron-folic acid tablets for 90+ days during pregnancy. Similarly, 19% of the population in Asia and 56% in Africa are food-insecure. Countries in Asia, like India and Bangladesh, and in Africa, like Ethiopia, Burkina Faso, Liberia, and South Sudan, etc., have some of the lowest coverage rates of one or more key interventions and practices addressing maternal and child malnutrition [3].

Maternal undernutrition is remarkably high in countries of South Asia, with Bangladesh reporting chronic energy deficiency among >30% of women of child-bearing age. Likewise, more than 16% of pregnant and lactating women are malnourished in Pakistan, 40% are anemic, and 12–22% are chronically energy deficient in Indonesia [1,4]. India also faces a high burden of maternal undernutrition, with more than 100 million adult women with a BMI < 18.5 kg/m^2^ [2]. Considering the magnitude of the problem, addressing maternal undernutrition has become a global priority. Sustainable Development Goal 3 has set a target of ending all forms of malnutrition and addressing the nutritional needs of adolescent girls and pregnant and lactating women by 2030 [5]. India has made strenuous efforts in this direction, and a decline in the prevalence of anemia and undernutrition (BMI < 18.5 kg/m^2^) among unmarried adolescents was achieved from 2005 to 2015 [6].

There is a need for a multi-pronged approach to address maternal undernutrition, given its multiple basic and underlying risk factors, as defined by the United Nations Children’s Fund (UNICEF) [7]. The risk factors associated with maternal undernutrition include low educational status, poverty, age between 15 and 34 years, living in rural areas, history of multiple pregnancies, parasitic intestinal infections, anemia, limited access to clean drinking water, limited access to nutritious and safe diet, poor marital conditions, lack of decision-making autonomy, and substance abuse [8,9,10,11,12]. The World Health Organization’s (WHO) latest guidelines on antenatal care (ANC) recommend fourteen evidence-based nutrition interventions towards a healthy pregnancy, including nutrition counselling, iron and folic acid (IFA), and calcium supplementation, etc. Additionally, emerging evidence recommends pregnant women consume balanced energy and protein supplements (BEP), like ready-to-cook or ready-to-eat food. BEP supplementation is a strategic approach to address chronic energy and protein deficiency among undernourished pregnant women [2].

A past review highlighted those interventions emphasizing counselling or delivery of information to women or their family members, such as consumption of iron-folic acid and calcium tablets by pregnant women for improved nutrition outcomes. Use of counselling material based on locally relevant information and engaging influencers, like mothers-in-law and husbands, are effective approaches to improve maternal nutrition outcomes. Furthermore, home visits and use of community forums for greater participation enhance access to services for pregnant women. However, digital interventions in silos are not proven to be an effective approach. The studies included in the review were limited and focused on either IFA or calcium supplementation or access to information concerning diet. Additionally, most of the available evidence reported in the review is from India, with limited data from other countries, like Pakistan, Indonesia, or Bangladesh [13].

To deliver evidence-based maternal-nutrition interventions, various implementation strategies have been initiated, yet maternal undernutrition continues to be a burden. Furthermore, it is imperative to understand and address maternal undernutrition due to its transgenerational impact, resulting in low-birth-weight babies, premature births, and chronic anemia [14]. Studies on the efficacy and effectiveness of these intervention strategies have been conducted in the past. However, to the best of our knowledge, there was lacking a review of the effectiveness of delivery mechanism(s) of such evidence-based maternal-nutrition interventions within the Southeast Asia region. Therefore, we conducted a scoping review to understand the various strategies to improve maternal nutrition, effective delivery mechanisms to improve maternal nutrition, and how interventions have improved coverage and uptake of services, as well as to document barriers and possible solutions for making such delivery mechanisms more effective.

## 2. Materials and Methods

This scoping review was developed and reported in accordance with Preferred Reporting Items for Systematic Review and Meta-analysis extension for scoping review (PRISMA-ScR) guidelines [15].

### 2.1. Research Question

The research question for this review was ‘What was the effectiveness of the different strategies for delivering facility- or community-based maternal-nutrition interventions/programs in countries of Southeast Asia?’. Additionally, ‘What were the outcomes, methodological approaches, and characteristics of the interventions/programs?’.

### 2.2. Information Sources and Search Strategy

A search strategy was developed by combining both MeSH terms and free-text words related to pregnancy and components of the nutrition program. An electronic search was performed in PubMed, Google Scholar, and the eLENA library, along with snowballing of relevant articles. The initial search was conducted on 08 November 2019; later, it was updated to 17 February 2021. We searched for articles that were published between 7 November 2010 and 7 November 2019 (last 10 years), later updated to 17 February 2021. The search was limited to the last 10+ years because of a lack of resources; furthermore, the authors perceived that interventions earlier than 10+ years might not be as relevant in the current scenario. The search was limited to studies published in English within 10+ years prior to the search date.

### 2.3. Eligibility Criteria

Peer-reviewed studies reporting strategies intended to improve maternal nutrition and/or increase coverage, uptake, and compliance with maternal nutrition services during pregnancy were included in this study. Studies of all design types, including randomized controlled trials (RCT), quasi-experimental trials, before-after studies, cross-sectional, case-control, etc., were included. All reviews, meta-analyses, letters, editorials, commentaries, conference papers, and reports were excluded. Studies reported from the WHO Southeast Asia region and Pakistan were considered for this review. The final search string used for PubMed can be found in the Appendix A.

### 2.4. Study-Selection Process

Screening of titles and abstracts was performed independently by two reviewers against the agreed inclusion and exclusion criteria. During each stage of screening, studies were taken forward only when both reviewers reached a mutual agreement for inclusion. A third reviewer arbitrated any disagreements between the reviewers to reach a consensus. Only those studies approved by both authors were included in the review, and the reason for the exclusion of studies was provided only at the full-text screening stage.

### 2.5. Data-Extraction Process

A standardized, pre-tested data-extraction form was developed. Data were extracted by one reviewer and verified by the other two reviewers. Categories under which data were extracted included (a) study characteristics, (b) methodological characteristics, (c) intervention strategies, and (d) targeted outcome of this review.

### 2.6. Collating, Summarizing, and Reporting the Results

A thematic narrative synthesis of included articles summarizing the effectiveness of each intervention strategy was carefully extracted from each included article. Intervention strategies from all the articles addressing each type of maternal-nutrition intervention were listed and coded.

## 3. Results

Search results: A total of 381 papers were identified from PubMed search, and eight from Google Scholar and reference snowballing. Of 381 papers, 315 were screened based on title and abstract (Figure 1). After the titles and abstracts were reviewed, 231 articles were excluded. During full-text screening, of the remaining 84 articles, 53 articles were excluded, leaving 31 articles for data extraction and quality appraisal. Finally, 22 studies were included for narrative synthesis.

### 3.1. Quality Appraisal

An appraisal was conducted for all papers included after full-text screening. The quality of evidence of each article included in the review was assessed independently by two reviewers using the Joanna Briggs Institute quality-appraisal tool [16]. The overall quality was good, as most of the articles satisfied the criteria in the checklist (Appendix A). There was no disagreement between the two reviewers concerning the inclusion of papers based on the quality checks.

### 3.2. Characteristics of Included Articles

Of all the 31 studies included in the review, only 22 have at least one outcome associated with delivery mechanisms; the study characteristics of all the 31 papers are summarized in Table 1 and Table 2 [17,18,19,20,21,22,23,24,25,26,27,28,29,30,31,32,33,34,35,36,37,38,39,40,41,42,43,44,45,46,47]. A total of 22 studies were ultimately included in this review for narrative synthesis [17,18,20,21,22,23,24,25,27,28,29,32,33,34,36,37,39,40,42,43,45,47]. Of these, twelve studies were RCTs, five were quasi-experimental trials, five were cross-sectional studies, three were mixed-method studies, three were secondary data analyses, one was a cohort study, and two were cross-sectional intervention/comparison studies. The majority (74%) of the studies was reported from India (12) and Bangladesh (11), three from Nepal, two from Pakistan (two others with India and Nepal), one from Indonesia, and two from Thailand (Myanmar). All identified studies were primarily quantitative, with a few studies reporting qualitative data as well.

### 3.3. Summary of Interventions

These studies include interventions of micronutrient supplementation (IFA, Calcium, and multiple micronutrients), deworming, nutrition education and counseling, BEP supplementation, and delivery mechanisms for implementing these interventions. The strategies adopted in these reviews were home visits, directly observed supplement intake, community mobilization, food and cash transfer, community volunteers, and peer-group education. Almost all interventions used community health workers as the primary human resource for delivering interventions to beneficiaries. Nearly two thirds of the studies showed improvement in the uptake/coverage/compliance with the intervention or services. Appendix A give an overview of the findings for each article included in the narrative synthesis.

Micronutrient supplementation (IFA, calcium, and multiple micronutrients)

Community health workers, female health workers, research assistants, and community volunteers are crucial for delivery of nutrition supplements. Home visits, in particular, for delivering nutrition supplementation by various grassroots-level workers were also reported. IFA supplementation programs in Pakistan and Nepal were delivered by community health workers, and coverage of IFA supplements significantly increased in the last ten years, from 23% to 80% in Nepal. However, in Pakistan, coverage remained stagnant, at 45%. Analysis also recorded IFA consumption with respect to the number of antenatal care (ANC) visits in Pakistan. IFA consumption of 45% was recorded among those who had at least one ANC visit, compared to 9.7% with no ANC visit; hence, ANC visit is a critical factor in this intervention [22].

IFA supplementation delivered by trained female health workers through a fortnightly home visit and a quarterly community-based group session reported that the proportion of intake of supplements was >76%, which was quite good, considering 65% had iron-deficiency anemia and low-birth-weight prevalence ranged from 18% to 40% [17]. An evaluation of the intervention package delivered through the trained community health workers and local health workers for improving perinatal and neonatal outcomes through community mobilization, education and home visits, use of information education and communication materials, videos, community health committees, and group meetings reported improved ANC services, including IFA and calcium supplementation [25].

Specially trained community health workers were reported to be the frontline facilitators delivering maternal nutrition services at the household, community, or facility level. MANOSHI, a community-based maternal, neonatal, and child health care service package among slum dwellers in Bangladesh delivered through female health workers through monthly home visits for IFA, calcium supplementation, and nutrition education, reported an increase in IFA consumption by 19% (*p* < 0.01) [28]. Similarly, the Government of Chhattisgarh, India, used community health workers, called mitanin (a local term for a close female friend), for improving the coverage of reproductive and child health services. Later, Nutrition Security Innovation, a project to inform families regarding their entitlement from the public distribution system, was integrated with this mitanin program to promote the appropriate complementary feeding through mitanins. An increased IFA consumption rate of 3% was reported, with a low occurrence of side effects [23].

Trained community health workers were recruited for delivery of ANC services, including nutrition-related services in Bangladesh through home visits or at community centers; their actions resulted in increased IFA consumption, with more than 82% of participants consuming more than 90 calcium tablets for more than three months. The study also reported a significant difference in the consumption of calcium tablets in relation to number of ANC visits; women who had four or more ANC checkups consumed more calcium tablets (*p* < 0.001) [39].

Directly observed consumption of supplements by the beneficiaries in the presence of community health workers was also associated with an increased consumption rate of multiple micronutrients (68%) and IFA (71%), indicating that successful distribution of supplements and direct observation of consumption of supplements improved adherence to consumption [18]. Additionally, a weekly home visit by community health workers to deliver micronutrient supplements was found to be associated with high compliance with supplements (median: 95, interquartile Range: 89.1, 98.4) [24]. Maternal nutrition knowledge and support from husbands were the key maternal and household factors associated with higher consumption of IFA and calcium tablets. Health service factors, like early and more prenatal care visits, ANC, and receipt of free supplements, also improved the consumption rate; for every home visit conducted by female health workers, four additional IFA and five additional calcium tablets were consumed by mothers. Combined exposure to these factors—knowledge, family support, and self-efficacy—was attributed to consumption of an additional 46 IFA and 53 calcium tablets, with 68% of pregnant women achieving minimum dietary diversity. Thus, regular home visits for delivery of micronutrient supplements at the doorsteps ensured that every pregnant woman received adequate supplementation [32]. Secondary analysis of an intervention study from India and Pakistan, where lipid-based micronutrient was provided through research assistants, showed a significant improvement in length-for-age Z-score (LAZ), weight-for-age Z-score, and reduction in low birth weight, as well as small-for-gestational-age incidence [47].

b.Nutrition education and counselling

Nutrition education and counselling provided by frontline health workers were informative and useful when provided by the mitanins in Chhattisgarh’s mitanin program. It was reported that a significant number of households among the intervention group adopted kitchen gardening (46.6% vs. 32.5%), and more than 98% reported having received information on the importance of IFA consumption. Involvement of community volunteers in family-level counselling for behavioral change and for delivery of nutrition services was also effective, and monthly nutrition and health education provided by Anganwadi workers (community-based frontline workers who promote child growth and development) during home visits or during village health and nutrition days was associated with 35% increased coverage in delivery of pregnancy care and nutrition information [23,27,41]. Studies from India and Bangladesh on health and nutrition programs showed that the trained community health workers were crucial in imparting nutrition education and counselling in these countries and were reported to have significantly improved the calcium-supplement consumption rate [20,39]. Monthly home visits by community health workers in Bangladesh to deliver nutrition education interventions resulted in 73.3% of pregnant women reporting having had visits from community health workers, and nearly three-quarters of pregnant women could name at least five food groups that should be consumed daily. However, large knowledge-to-practice gaps were observed in terms of food consumption [32]. A similar study evaluated the effect of providing intensified nutrition-focused maternal, neonatal, and child health intervention compared to standard interventions. More than 96% of pregnant women in the intensified focused nutrition group received IFA and calcium tablets, and 40–50% of all mothers reported exposure to video shows. Proportion of IFA and calcium intake, number and quantity of food groups consumed, and daily intake of macronutrients improved substantially in the intensified focused nutrition group, as compared to the control group [33].

The participatory learning and action approach engages communities in learning and participation, identifying needs, planning, nutrition education, and counselling, and encouraging problem-solving. Studies were designed to assess the effects of pregnancy-focused nutrition interventions through the participatory learning and action approach, combined with the transfer of food or cash in intra-household food allocation, dietary adequacy, and maternal nutritional status in Nepal. Monthly group sessions were conducted in all the arms. The participatory approach used picture cards and stories for group discussions. A structured manual was prepared for nutrition mobilizers to conduct home visits and nutrition counselling to improve nutrition practice. The participatory learning and action approach with cash or food transfer reported a higher attendance rate at monthly meetings (80%) compared to the approach alone without cash or food transfer. The participatory learning and action approach with food transfer significantly improved the birth weight of the fetus and equity in energy allocation among pregnant women. It also improved dietary diversity and adequacy, as well as supplement consumption [37,40]. Conditional cash transfer of INR 5000 (Indian Rupee) delivered in four installments (1500 at end of second trimester, 1500 at third month after delivery, 1000 at the sixth month, and 1000 at the ninth month) was reported from India. Each installment was conditional on the uptake of certain health care services and outcomes. Compliance with conditions, such as consumption of IFA tablets; counselling during pregnancy; and the recommended immunization for bacillus Calmette-Guerin, polio, and diphtheria-pertussis-tetanus were observed to be significantly higher (unadjusted *p* < 0.01) among those receiving conditional cash transfer. Conditional cash transfer was also observed to be associated with an increased likelihood of pregnancy registration and receiving ANC services and IFA tablets from female health workers [34]. Counselling on maternal nutrition through home visits and participatory women’s group meetings facilitated by community health workers at least twice a month was associated with a significant increase in the odds of pregnant women achieving minimum dietary diversity in intervention clusters (adjusted odds ratio 1.40; 95% CI 1.03 to 1.90, *p* = 0.0311) [36].

For the promotion of maternal, infant, and young-child nutrition, the components of behavior-change communication were integrated into agricultural extension programs on nutrition by developing cost-effective, video-based intervention. This showed that intake of IFA tablets was 86% among those who had received information, and several female health workers requested that the videos be disseminated during village health and nutrition days and during the monthly health and nutrition fairs. Female health workers also reported that the videos further helped in serving the community well. Interviews with various health workers, self-help groups, volunteers, and beneficiaries reported an increase in uptake and awareness of government services for improving nutrition [29].

c.Balanced energy-protein supplementation

Although evidence to evaluate potential long-term outcomes is not adequate, a few studies reported that supplementation of BEP diet in undernourished pregnant women facilitated gestational weight gain and improved fetal outcomes in terms of reducing the risk of stillbirth, low-birth-weight infants, and small-for-gestational-age babies. In Nepal, the provision of 10 kg per month of a fortified BEP supplement of wheat-soya blended flour with 10% added sugar, called super cereal, delivered by community health workers, along with the participatory learning and action approach, significantly improved the birth weight of newborns and equity in energy allocation among pregnant women [37,40]. Similarly, in Bangladesh, a locally produced food-based BEP supplementation for undernourished pregnant women delivered along with nutrition education and counselling and regularly monitored by community nutrition volunteers either at the designated center in the community or through home visits showed a 98% compliance rate, with almost all women consuming the full supplement daily [43]. However, evaluation of supplementary food programs delivered in India through the Integrated Child Development Service Scheme by Anganwadi workers reported that only 20.5% of women received supplementary food during their pregnancies due to a weak procurement system and poor quality of food products [20]. Another study from India and Pakistan provided additional protein-energy supplements to pregnant women whose BMI was <20 kg/m^2^ only in the interventional groups. LAZ, weight-for-age z score, low birth weight, and small-for-gestational-age were much improved in the interventional groups as compared to the control group. Prevalence of newborn stunting (LAZ <−2) was 18% (Intervention 1: 10% & Intervention 2: 13%), wasting (weight-to-length ratio for age < −2) 42% (32% & 37%), preterm deliveries 12% (12% &8.5%), low birth weight 34% (28% & 29%) and small for gestational age 49% (36% & 44%) in the control group, as compared to the intervention groups [47].

d.Barriers affecting coverage, uptake, and compliance of program delivery

Public health interventions are complex, and outcomes are unpredictable. Factors affecting the success of the programs are multidimensional. A few barriers to interventions were reported in some of the studies identified for this scoping review. An evaluation of the implementation of maternal nutrition programs in India reported that only 27.6% consumed at least 100 IFA tablets or syrup during their most recent pregnancy, only 4% took deworming medicine during pregnancy, 20.5% received supplementary food, and 10.9% received nutrition and health education. Similarly, the report recorded a meager utilization rate of the nutrition services provided at anganwadi centers and food-fortification projects implemented through the public distribution system. Inadequate and irregular supply of supplements, substandard quality of the food provided, misconceptions about intake of certain food items during pregnancy, and sharing of take-home rations with the household members were a few barriers that hindered program effectiveness [20]. Other factors that affected the delivery and uptake of interventions were resource shortages, socioenvironmental issues, including poverty, lack of awareness, and discrimination based on socioeconomic status in the community [21].

Certain policy-related barriers reported in different studies were low prioritization of maternal-nutrition intervention by policymakers and lack of proper thematic knowledge of the program reference. All these barriers were found to be associated with inefficient program management within the overall health system. One study also reported a low consumption rate of deworming pills (4%), despite receiving the targeted intervention, due to weak monitoring, evaluation, and not prioritizing time-bound targets [21]. Barriers to IFA consumption were reported to be a lack of appropriate need forecasting, delay in supply, inconsistent training on IFA counselling/distribution, low health literacy, unplanned pregnancy, no or late pregnancy registration, limited intervention resources, and misconceptions [42,45].

## 4. Discussion

This review is different from other reviews conducted on maternal-nutrition intervention in several ways. The objective of this review was not to identify evidence-based maternal-nutrition interventions but to document intervention strategies that have successfully improved coverage and uptake of maternal nutrition services and programs. Review of implementation of various evidence-based interventions to improve maternal nutrition and birth outcomes indicates that community health workers, home visits, directly observed nutrition supplementation, community mobilization, and social marketing approaches for delivery of interventions like IFA supplementation, deworming, BEP supplementation, and nutrition education and counselling provide better results in terms of coverage, compliance with services and uptake of these interventions. The present review qualitatively narrates various intervention strategies adopted to deliver programs to improve maternal nutrition and birth outcomes.

### 4.1. Effect of Home Visits by CHW on Compliance with Nutrition Interventions

Maternal nutrition status is crucial for fetal growth and development; nutrition supplements are effective in improving maternal and fetal outcomes and have been routinely administered to pregnant women [48,49,50]. The review found that auxiliary nurse midwives, female health workers, community health workers, Anganwadi workers, accredited social health activists, and community volunteers all play a crucial role in delivering public health interventions either at a designated facility or community centers, on a designated day or through house visits. Reported outcomes of various studies showed an increase in supplement consumption rate when nutrition services were delivered by community health workers through home visits [17,18,22,23]. These findings support other reviews reporting on the delivery of services through home visits by community health workers. Thus, by ensuring universal health care with strengthened primary health services, the prevalence of anemia and undernutrition could be reduced [6]. Sanghvi et al. 2016 [51] examined evidence of IFA supplementation and the effectiveness of intervention trials in a large-scale program and found that the active involvement of village health volunteers through home visits helps to improve compliance and ANC visits.

Mason et al. 2012 [52] assessed the current scenario of IFA supplementation in India and reported that community health workers and auxiliary nurse midwives were the driving force for delivery of supplementation and observed an improvement in the uptake of services and intervention coverage. A recent population-based intervention study conducted by Edmond et al. (2018) [53] in Afghanistan evaluated the effectiveness of a program of home visits by trained community health workers and reported an increase in institutional delivery of 8.2% in the intervention group, as compared to 6.3% in the control group. There was also an increase in ANC visits of 3.4% in the intervention group, whereas a decline of 1% was observed in the control group. The findings show that home visits by community health workers improved care-seeking behavior and knowledge among pregnant women. Overall, home visits by community health workers increased coverage and compliance with the consumption of micronutrient supplements and uptake of ANC services.

Many studies lack data on the direct effect of health workers on the uptake of services/compliance (as revealed through association/regression/attribution) [12,18,19,23,24,26,29,32,33,34,35,37,38,42,43]. These studies compared the effect of the intervention (as a whole) on the outcomes without delineating the isolated effects of health workers/delivery mechanisms. However, some studies reported on the impact of CHW on health outcomes. In the MANOSHI program, CHW/trained providers/medically trained providers who paid monthly antenatal and postnatal visits reported greater effect of CHW on improving the uptake of four or more ANC visits and quality ANC visits than the routine services [28]. Frontline workers, like ASHA/AWW who were of the same caste as the head of the household and were living in the same catchment area of pregnant women, had higher odds of providing immunization services to children under 5 years. and pregnancy-related counselling, respectively [27]. CHWs trained on additional curriculum of newborn care and with improved counselling skills enhanced maternal and newborn care practices significantly compared to those who did not receive additional training and skill development [25]. However, CHW visits in addition to existing AWW were not found to significantly affect child health outcomes [36].

### 4.2. Effects of Nutrition Education and Counselling on Compliance and Uptake of Nutrition Services or Intervention

Existing evidence suggests that nutrition education and counselling is an effective strategy, but the impact is more significant and substantial among undernourished populations of low- and middle-income countries when accompanied by nutritional services, like provision of free micronutrient supplements, ANC care, periodic reinforcements, and food supplementation [54,55]. Nutrition education and counselling is the primary responsibility of community health workers and auxiliary nurse midwives, and every nutrition intervention and program has a component of nutrition education and counselling either at the individual, household, community or facility level. Delivery of nutrition education by trained community health workers or female health workers was reported to be associated with good nutrition knowledge, greater support from the husband, higher consumption of supplements, and achievement of minimum dietary diversity [20,27,32,39]. Home-based counselling, group meetings, and community events provided by frontline workers were also reported to be informative and useful [41]. A review by Girard et al. 2012 [51] reported that nutrition education and counselling targeting maternal diet and supplement intakes during pregnancy helped to improve dietary patterns and adherence to a healthy diet. Studies have also reported that nutrition education and counselling increased compliance with micronutrient supplementation and uptake and utilization of ANC services, which is in line with the findings of the present study. Another review [56] reported that large-scale programs involving nutrition education and counselling usually delivered by community health workers and a program in Bangladesh on nutritional counselling for behavior change succeeded in creating awareness and attention towards undernutrition and in designing a sustainable nutrition program in the country. Similarly, a review by Vaivada et al. [57] reported that nutrition education and counselling for pregnant women in undernourished populations was effective in improving birth outcomes and home visits by specially trained community health workers for counselling and IFA supplement and facilitates decision making concerning IFA consumption [58].

Participatory learning and action (PLA) in combination with cash or food transfer led to higher attendance rates at monthly meetings, improved birth weight, and equity in energy allocation among pregnant women, while PLA with cash transfer improved adequate dietary diversity and supplement consumption [37,40]. Community interventions through home visits and PLA by local community-based workers significantly achieved adequate dietary diversity among pregnant women [36,42]. In concurrence with this rapid review, a non-RCT by Gope et al. 2019 [59] weighed the effects of two strategies concerning monthly PLA meetings with women’s groups followed by home visits and crèches combined with monthly PLA meetings and home visits. Trained community health workers facilitated the implementation, and the study reported that 65–72% of mothers participated in the meetings, and an increase in the uptake of services, early initiation of breastfeeding, adequate dietary diversity among mothers, and consumption of iron-rich foods by children were recorded. PLA meetings, along with home visits, reduced undernutrition among children, suggesting the potential benefit of scaling up the intervention through accredited social health activists and their facilitators.

### 4.3. Effects of Food or Cash Transfer on Compliance and Uptake of Nutrition Services

Take-home rations and supply of fortified food through the public distribution system was one of the interventions used to address maternal undernutrition and low birth weight, [21] but a study from India revealed that 35% of the sample households received food supplements from *Anganwadi* workers and accredited social health activists, and only 20.5% women reported to have received supplementary food during their pregnancy through the integrated child development service scheme (ICDS) program [20,27]. Inversely, in Bangladesh, the delivery of locally produced prenatal food-based supplements to pregnant women led to a high compliance rate, with almost all participants consuming the full supplement on a daily basis [43]. Similarly, in Nepal, food and cash transfer with PLA led a high participation rate in program activities and significant improvement in birth weight of newborns, equity in energy allocation, and adequate dietary diversity [37,40]. Earlier reviews found conditional cash transfer to be a better intervention strategy than direct distribution of food supplements. Mexico and Brazil succeeded in conditional cash transfer programs, and this approach is now being endorsed and implemented in other low- and middle-income countries [54,56].

Another study from Mexico [60] on the nutritional impact of a large-scale, incentive-based development program (PROGRESA), which includes the provision of micronutrient fortified food supplements to eligible women and cash transfers of USD 25 per family, with the condition of complying with specific healthcare appointments, including a mandatory session on nutrition and health education, showed an overall improvement in the consumption of nutrient supplements, improvement in height, and reduction in anemic cases. This indicates the potential benefit of the program for improving adherence to the intervention.

In India, the Janani Suraksha Yojana (JSY) program provided cash benefits to pregnant women from low-income households to promote institutional delivery, which led to an increase in the utilization of ANC and skilled delivery. This, in turn, led to a decline in perinatal mortality. Conversely, the provision of conditional monthly food rations and micronutrient-fortified individual rations for daily consumption improved the participation rate by more than 95% for monthly group sessions. High attendance and participation rates indicate that delivery of targeted nutrition intervention through conditional food ration facilitates program effectiveness in achieving its desired outcomes [61,62].

### 4.4. Effect of Community-Level Events, Social Marketing Campaigns or Group Sessions on Compliance and Uptake of Nutrition Services

Village health and nutrition day (VHND) in India, maternal and child health weeks in Nigeria, and enhanced outreach strategies or community health days in Ethiopia are examples of campaign-based approaches for delivery of nutrition interventions. VHND, a monthly event usually organized at Anganwadi centers, provides nutritional services, such as nutrition education and counselling and fortified take-home food rations to pregnant women, lactating mothers, and children. In Ethiopia and Nigeria, pregnant women and lactating mothers are screened to determine targeted beneficiaries for food supplementation through health days or health-week campaigns, as the majority of the women is reported to miss ANC visits [54]. Similarly, several reports in this review also discussed social campaigns, community events, or designated days for delivery of interventions. VHND is a beneficial platform for reaching those who did not receive services through other platforms, utilization of information, education and communication materials for conducting community group meetings, home-based sessions for delivery of nutrition education messages, delivery of supplements and problem-solving through peer group discussions, etc. [20,25,42].

Community-level campaigns were found to be effective in creating awareness and promoting the intake of supplements. Social marketing and community-based education sessions using information, education, and communication material before distribution of supplements were found to be effective in the uptake and usage of supplements [19]. Similar interventions reported that the involvement of community health workers in community mobilization activities and home visits helps in tracking uptake and utilization of nutritional services and identifying defaulters [63]. A study from Africa reported an increase in the utilization rate of healthcare-related services following intensive and exhaustive community-mobilization and advocacy activities [64]. Similarly, a study from Vietnam promoting IFA consumption reported community mobilization and social marketing as an effective intervention strategy for promotion of uptake of services and intervention coverage. The study also reported a significant change in the knowledge, attitude, and practice of the targeted beneficiaries related to health and nutrition [65].

### 4.5. Strengths and Limitations

We have identified a wider range of maternal nutrition programs and their implementation through a stringent process to include them in this review. However, only PubMed and Google Scholar were searched to identify articles, as there was a lack of time and resources. Owing to this, there could be a greater chance of missing out on other relevant studies not indexed in PubMed or Google Scholar. Additionally, meta-analysis was not performed in the present paper due to varied outcome indicators and non-uniformity in the presentation of results across studies. However, the studies included in the review present a diverse range of strategies to improve maternal nutrition in many countries, including India, Pakistan, Indonesia, etc., compared to the previous reviews. Furthermore, many aspects of maternal nutrition programs were discussed, including strategies, effectiveness to improve maternal and child health outcomes, and delivery mechanisms. This makes this paper relevant for policy makers, public health specialists, and academicians.

## 5. Conclusions

Specific interventions targeting maternal nutrition have proven to be effective in improving pregnancy outcomes. However, the proportion of people who benefitted from these interventions in South-East Asian countries, with a high prevalence of anemia and underweight women and low rates of IFA consumption, [66] needs to be explored. Implementing public health programs, small or large, is complex and requires an understanding of cultural diversity, the public health system, and population characteristics, and program implementers should examine the benefits of effective intervention strategies for context-specific and effective program design.

In this review, intervention strategies, such as home visits, nutrition education, and counselling through different means, including video-based sessions, free supplementation of IFA/calcium tablets, monitoring of ANC visits, and community mobilization and meetings, were found to be associated with higher intervention coverage, increased compliance with intervention, and uptake of services. However, intervention strategies such as food rations and food transfer showed varied effects. Studies from India reported that the quality of food supplied under food-transfer programs was not suitable for consumption, which inversely affected program coverage. Cash transfer, on the other hand, led to improved compliance with interventions, as reported by studies from Nepal, although evidence was not definitive and needs further evaluation.

The overall findings from this scoping review were that strengthening of ANC at the community or facility level to promote ANC visits, frequent home visits, building capacity of community health workers, promoting participatory community mobilization, nutrition campaigns, conditional cash transfer, provision of food supplements, and improving quality of food supplements improve coverage, compliance, and uptake of services. Furthermore, there is a need to improve quality and regular supply and address the shortage of food supplements, with continuous monitoring. Solutions to address barriers related to program delivery are needed to further improve effectiveness. Besides, the socioenvironmental factors such as poverty, lack of awareness, and misconceptions in the community require immediate action. The findings of this review will help decision makers and program implementers who seek to understand the complexity of implementing such evidence-based interventions addressing maternal undernutrition and birth outcomes for effective program design and improvement of program efficiency and effectiveness.

## Figures and Tables

**Figure 1 ijerph-18-13292-f001:**
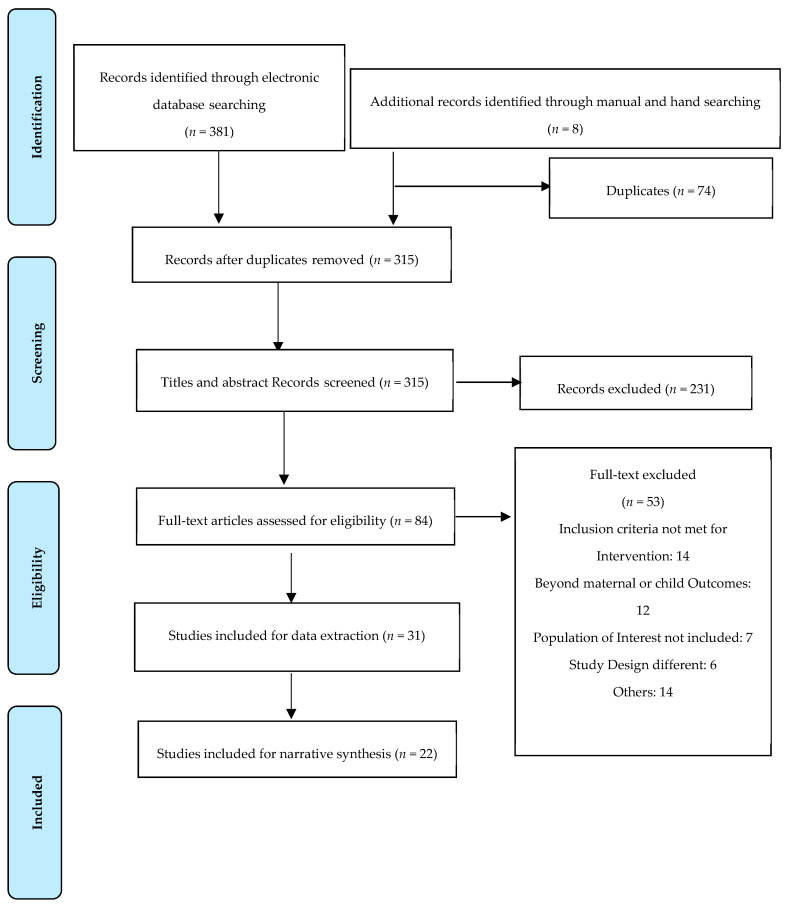
A PRISMA flow chart illustrating search and screening results.

**Table 1 ijerph-18-13292-t001:** Characteristics of the included studies (*n* = 16).

Authors	Study Design	Sample Size	Study Population	Location of the Study	Intervention/ Comparison	Study Outcomes	Delivery Mechanism	Change in Coverage/Uptake
Bhutta, et al. (2009) [17]	Cluster RCT	I: 1148 C: 1230	Pregnant women	Pakistan (rural and urban)	I: Multiple micronutrients C: IFA only	Increased maternal monthly weight gain and reduced LBW prevalence in the intervention group; no significant improvement in iron status of women, but the control group had a higher prevalence of subclinical zinc deficiency.	Trained female CHW visited fortnightly	No significant difference in uptake of ANC visits or consumption of IFA tablets
Sunawang, et al. (2009) [18]	Cluster RCT	I: 432 C: 411	Pregnant women	Indonesia	I: Multiple micronutrients C: IFA only	No statistical difference in the birth parameters of child (birth weight, length, head circumference), pregnancy outcome (miscarriage, stillbirth, or neonatal death), or maternal hemoglobin or serum levels of zinc, retinol and ferritin, and urinary iodine.	Field workers of the project visited daily (except Sunday)	High rate of adherence to supplementation (uptake of IFA or MNP) observed in both groups (no statistical difference) due to home visits and monitoring by field workers
Rah, et al. (2011) [19]	Cross-sectional assessment in intervention and control areas	I: 358 C: 361	Lactating mothers	Bangladesh	I: Micronutrient powder + IFA tablets + fortified food ration + social marketing campaign + education sessions C: IFA tablets + fortified food ration	No statistical difference in mean weight, height, BMI, hemoglobin levels, and anemia prevalence between intervention or control group, except proportion of thinness and decreased anemia in women consuming at least 75% sachets, compared to those consuming <75% sachets within the intervention group.	MNP distributors, health and nutrition staff of local NGOs with doctors, midwives, nutritionists, and volunteers	Increased uptake of MNP was reported (compliance)
Ramakrishnan, et al. (2012) [20]	Cross-sectional qualitative research	KII: 31 FGD: 35 IDI: 15	Program officials, health workers (ASHA/ANM/AWW/Doctors) at the central and peripheral level, community leaders, and volunteers. Women 18–45 years. with children ≤3 years.	Tamil Nadu and Uttar Pradesh, India	-	Only 27.6% consumed ≥100 IFA tablets during their most recent pregnancy, 4% took deworming medicine during pregnancy, 20.5% received supplementary food, and 10.9% received nutrition and health education. Barriers were lack of supply of supplements, quality of food, misconceptions concerning intake of certain food products during pregnancy, and in case of take-home ration, sharing of food with the household.	Frontline workers (ASHA, AWW, ANM)	Effective counselling by health workers and targeted media campaigns improve uptake (compliance)
Noznesky, et al. (2012) [21]	Qualitative research using KII	KII: 48	Policy makers, program managers, service providers	Bihar, India		Only 4% consumed deworming pill during their most recent pregnancy, and 0.6% utilized *Anganwadi* services. Barriers were resource shortages, poverty, lack of awareness, discrimination based on socioeconomic status, as well as policy-related barriers, such as lack of focus on maternal nutrition, knowledge level of the program implementers about maternal undernutrition, and a faulty program management.	Frontline workers (ASHA, AWW, ANM)	Improved essential inputs, management systems, and reduced gender or caste discrimination
Nisar, et al. (2014) [22]	Secondary data analysis: NDHS (2006/2011) and PDHS (2006–07/2012–13)	Pooled NDHS: 8196 mothers Pooled PDHS: 13034 mothers	Mothers	Nepal and Pakistan	I: Any use of IFA or >90 IFA supplements C: No use of IFA	AR of early neonatal deaths was significantly reduced, by 51% in Nepal and 23% in Pakistan, with any use of IFA compared to none. When IFA started at or before the fifth month of pregnancy, the AR of early neonatal mortality was significantly reduced, by 53% in Nepal and 28% in Pakistan, compared to no IFA. When >90 IFA supplements were used and started at or before the fifth month, AR of early neonatal deaths significantly reduced, by 57% in Nepal and 45% in Pakistan.	Public-sector facilities and CHW	Training CHW, making IFA supplements available for free, and increasing demands through awareness-promotion programs
Vir, et al. (2014) [23]	Quasi experimental (mixed methods assessment)	I: 1825 C: 1801	Mothers of children <3 years	Chhattisgarh, India	I *: NSI + *Mitanin* program C: Only *Mitanin* program	No significant difference in the nutritional status of children between two groups; more households in the intervention group than in the control group had kitchen gardens (46.6% vs. 32.5%).	Female CHWs (*Mitanin*)	No significant change in the uptake of IFA tablets but significant improvement in coverage of three ANC visits and ANC within first trimester
Gernand, et al. (2015) [24]	Double-blind, cluster-RCT	I: 264 C: 236	Pregnant women	Bangladesh (rural)	I: Multiple micronutrient powder C: IFA	No difference in maternal plasma levels of hPL o PGH or cord plasma levels of insulin, IGF-1, or IGFBP-1 between two groups; however, higher cord insulin concentration was in women who were short and higher hPL was found in women carrying female fetuses.	Local field workers visited weekly	High compliance with both MNP and IFA noted in the study
Memon ^β^, et al. (2015) [25]	Exploratory quasi-experimental design	I: Pre (*n* = 322) and Post (*n* = 316) C: Pre (*n* = 386) and Post (*n* = 361)	Pregnant women	Northern Pakistan	I: Community mobilization + education on MNHC through CHC and group sessions + Routine health services C: Routine health services	Improvement in ANC, TT vaccination, institutional delivery rate, cord application, delayed bathing, colostrum feeding, early initiation of breastfeeding (<1 h of birth) (*p* < 0.001), and reduction in perinatal and neonatal mortality rates (*p* < 0.05) in the intervention group.	Female health workers and CHW conducted monthly household visits, one-to-one sessions, and video sessions	Increased uptake of ANC visits and care
Sablok, et al. (2015) [26]	RCT	I: 120 C: 60	Pregnant women	Delhi, India	I: Vit D Supplementation C: No supplementation of Vit D	Intervention group had lower incidence of preterm labour (*p* = 0.02); higher number of newborns to mothers in the control group had lower Vit D levels (<25 nmol/L) (*p* < 0.001) and lower mean birth weight; and higher proportion of SGA in control group (*p* = 0.04).	Unclear (Department of Obstetrics and Gynecology, Tertiary hospital, Delhi)	Not reported
Kosec, et al. (2015) [27]	Secondary data analysis of DLHS and facility workers Surveys 2012	6002 households in 400 villages	Household and frontline workers (ASHA, AWW)	Bihar, India	-	Monetary incentives for AWW are strong predictor of receipt of immunization services (0–2 years) and households receiving general nutrition information.	ASHA and AWW delivering routine services	Incentivizing frontline workers and improving performance increase service uptake by households
Jolly, et al. (2016) [28]	Cross-sectional comparative study	I: 607 C: 599	Married women	Bangladesh (Urban slums)	I: MANOSHI ^¶^ program C: Without MANOSHI services	Increased odds of improved maternal-health service indicators (4 or more ANC visits, receipt of IFA and TT injection, PNC within 48 h of birth, and institutional delivery rate) in the intervention group (*p* < 0.05).	Female CHW doing household visits	Uptake of four or more ANC checkups, institutional delivery, skilled assisted delivery, and PNC increased in the intervention group
Kadiyala, et al. (2016) [29]	A case study of digital green approach (qualitative assessment)	** I: IDI: 72 SSI: 73 KII: 6	SHG members (PLW, CFM, mothers of adolescent girls, and other women); MIL, husbands, FLW, key stakeholders, protagonists, FLPP ^⁑^	Odisha, India	I: MIYCN BCC + Digital green approach to agriculture extension	Intervention well received by rural communities and viewed as complementary to routine services; intervention was perceived as a credible source of information related to health and nutrition.	CHWs	Participatory, dialogue-based interventions with women’s groups, as well as video education, improved maternal and childcare services
Mridha, et al. (2016) [30]	Researcher-blind, longitudinal, cluster-randomized effectiveness trial	I: 1047 C: 2964	Pregnant women ≤ 20 gestational weeks	Bangladesh (rural)	I: -Mothers given LNS-PLs C: Mother given IFA	Increased mean birth weight, WAZ, birth length, LAZ, head circumference, HCZ, BMIZ in the children born to mothers in the intervention group	NGO staff	Adherence to IFA was more than the intervention
Rahman, et al. (2016) [31]	Quasi-experimental study	I: Pre: 4800; Post: 2400 C: Pre: 2400; Post: 1200	Mothers and children	Bangladesh (rural)	I: Intensive maternal and newborn care services ^§^ C: Essential health care services	Increased number of ANC visits (≥4), skilled birth attendance at dlivery, and PNC visits (≥3 visits); reduction in ANC and PNC complications; mother’s knowledge of breastfeeding initiation and initiation of breastfeeding within an hour of birth increased in the intervention group.	CHW	Uptake of family planning methods, four or more ANC visits improved in the intervention group
Nguyen, et al. (2017) [32]	Cross-sectional study	Pregnant women (*n* = 600); Recently delivered (*n* = 2000)	Pregnant women and recently delivered	Bangladesh	I: Standard nutrition intervention of nutrition education + IFA + and calcium supplementation + deworming + IYCF counseling	Good nutrition knowledge, women’s self-efficacy, perception of enabling social norms, high husband’s support, early and more prenatal visits, provision of free supplements improve maternal nutrition practices (IFA and calcium intake and diverse diets).	Frontline health workers conducted monthly home visits	Observed increase in the uptake of IFA/calcium tablets and four or more prenatal visits

Abbreviations: ANC: antenatal Care; ANM: auxiliar nurse midwife; ASHA: accredited social health activist; AWW: anganwadi workers; AR: adjusted risk; BMI: body mass index; BCC: behavior-change communication; BMIZ: body-mass-index for-age Z scores; C: control group; CHC: community health committee; CHW: community health workers; CFM: complementary feeding mothers; DLHS: district-level household survey; FGD: focus-group discussion; FLW: frontline health workers; FLPP: field-level program personnel; hPL: human placental lactogen; HCZ: head-circumference-for-age Z score; IDI: in-depth interview; IGF: insulin-like growth factor; IGFBP-1: insulin-like growth factor binding protein-1; IFA: iron folic acid; I: intervention group; IYCF: infant and young-child feeding; KII: key informant interview; LAZ: Length-for-age Z scores; LBW: low birth weight; LNS-PLs: lipid-based nutrient supplements for pregnant and lactating women; MNP: micronutrient powder; MNHC: maternal newborn health care; MIL: mothers-in-law; MIYCN: maternal, infant, and young-child nutrition; NDHS: Nepal Demographic Health Survey; NGO: non-governmental organization; NSI: nutrition security innovation; PGH: placental growth hormone; PDHS: Pakistan Demographic Health Surveys; PLW: pregnant and lactating women; PNC: post-natal care; RCT: randomized controlled trial; SHG: self-help group, SGA: small for gestational age; SSI: semi-structured interview; TT: tetanus toxoid; Vit D: Vitamin D; WAZ: weight-for-age Z scores. ^β^ The intervention was delivered to 16802 households covering 3200 pregnant women, and there were 18659 households in the control area. The MNCH included awareness creation about positive maternal and newborn health care practices at the household level, such as the importance of seeking antenatal care, adequate nutrition during pregnancy and lactation, skilled birth attendance (antenatal care, early initiation of breastfeeding, delayed bathing, and recognition of danger signs that warrant early referrals), and practices promoted through community mobilization and education strategies that included formation of community health committees and group sessions using flip charts and videos. * NSI included intensive behavioral-change communication to promote appropriate complementary feeding practices, exclusive breastfeeding, establishment of kitchen gardens, and informing the community of their entitlements to subsidized food items through the state-modified public distribution system (PDS), which is a food-security program and is expected to supply a minimum food basket of cereals, sugar, and kerosene cooking fuel at a subsidized cost. Mitanin program included mitanin as counselors for families with either pregnant women or children under three years of age to improve coverage of maternal and child health services. ^¶^ MANOSHI is a community-based maternal, newborn, and child health care service package utilizing female CHWs (paid renumeration) to promote family-planning methods and provide door-to-door antenatal and postnatal care checkups to women. ** Intervention was conducted across 30 villages. ^⁑^ Field-level program personnel included community service providers and community resource persons. Protagonists included persons who featured videos on mother and infant young-child nutrition. Key stakeholders included persons from partner organization. ^§^ Intensive maternal and newborn care services included formation of village-based maternal neonatal child health committees, training of traditional birth attendants on safe deliveriy, promotion of antenatal and postnatal care practices, tetanus toxoid injection, birth planning, counselling and communication strategy, adequate maternal nutrition, effective referral system, newborn care practices, complementary feeding, delayed bathing, and increased health-workers attendance at delivery.

**Table 2 ijerph-18-13292-t002:** Characteristics of the included studies (*n* = 15).

Authors	Study Design	Sample Size	Study Population	Location of the Study	Intervention/ Comparison	Study Outcomes	Delivery Mechanism	Change in Coverage/Uptake
Nguyen, et al. (2017) [33]	Cluster RCT with cross-sectional baseline (2015) and endline (2016) survey	I: PW: 300 RDW: 1000 C: PW: 300 RDW: 1000	Pregnant women and recently delivered women	Bangladesh	I: nutrition-focused ^¶^ MNCH intervention C: standard MNCH intervention	Increase in consumption of IFA and calcium supplements, ≥5 food groups, and most macro and micronutrients; increase in individual food groups consumed among women; increase in EBF by women	Salaried health workers and community health volunteers conducted monthly home visits	Increased probability of early ANC visits and receipt of free iron and calcium tablets in the intervention group
Raghunathan, et al. (2017) [34]	Cross-sectional study	I: 534 C: 627	Pregnant and lactating women	Odisha, India	I: women who received money under CCT scheme C: women who did not receive money under CCT scheme	Increase in likelihood of pregnancy registration, receiving ANC services (5 pp), and IFA tablets (10 pp) and a decline of 0.84 on the household food insecurity assessment scale	DBT by the government and essential nutrition intervention by AWW and ASHA	CCT scheme increased the coverage of ANC services, IFA consumption, and pregnancy registration
Dewey, et al. (2017) [35]	Researcher-blind, longitudinal, cluster-randomized effectiveness trial	^α^ IFA-MNP = 1052; IFA-LNS = 930; LNS-LNS = 1047; Control (IFA) = 982	Pregnant women at ≤20 gestational age and children	Bangladesh	^α^ I: IFA-MNP: mother given IFA and child given MNP; IFA-LNS: mother given IFA and child LNS; LNS-LNS: mother and child both given LNS; C: IFA-control: mother given IFA and child given nothing	LNS-LNS group had significantly higher LAZ (+0.13 compared with the IFA-MNP group) and head circumference (+0.15 z score compared with the IFA-Control group); stunting prevalence (LAZ < −2) was lower in the LNS-LNS group at 18 months than in the IFA-MNP group (OR: 0.70; 95% CI: 0.53, 0.92), but the difference diminished by 24 months (OR: 0.81; 95% CI: 0.63, 1.04)	NGO staff	Adherence to the interventions was reported but not to other services, which was higher in LNS-LNS and IFA-LNS than LNS-MNP
Nair, et al. (2017) [36]	Cluster RCT	I: PW: 2805; Infants: 1460 C: PW: 2952; Infants: 1541	Pregnant women and infants	Jharkhand and Odisha, India	I: Single home visit during 3rd trimester, monthly home visit to children < 2 years for counseling and growth promotion, 2–3 participatory meetings with local women’s groups	No significant effect on EBF, timely initiation of complementary feeding, morbidity, appropriate home care, or care-seeking during childhood illnesses of the intervention; more pregnant women and children attained MDD, more mothers washed their hands before feeding children, fewer children were underweight at 18 months, and fewer infants died	Community-based incentivized volunteers	No significant change in the uptake of maternal or childcare services
Harris-Fry, et al. (2018) [37]	Cluster RCT	I: PLA: 154 PLA + Cash: 283 PLA + food: 218 C: 150	Pregnant women	Nepal	* I: Women’s groups practicing PLA, PLA women’s groups with a monthly unconditional food transfer, and PLA women’s groups with a monthly unconditional cash transfer; C: usual government services	All of intervention groups had increased consumption of IFA supplements, MUAC measurements, and intrahousehold allocation of some animal-source foods; however, RDEARs between pregnant women and their mothers-in-law were higher in PLA + food arm, and dietary diversity was 0.4 food groups higher in PLA + cash arm than control arm	Government-incentivized female community health volunteers and nutrition mobilizers	Significant uptake of IFA supplements in the intervention groups
Hashmi, et al. (2018) [38]	Convergent parallel mixed-method design	Cross sectional survey = 388 PW FGD = 11 women IDI = 4 midwives	Pregnant women	Thailand	-	A high proportion of women had limited knowledge of and poor dietary practices. Sweetened-drink consumption in the last 24 h, as well as being non-teenaged multigravida woman, significantly associated with high BMI compared to normal BMI	-	Proportion of first antenatal care visit higher for the first trimester than in the second or third trimesters
Khanam, et al. (2018) [39]	Retrospective cohort design	Case: Women who had PIH Control: Women did not develop PIH	Pregnant women	Bangladesh	I: MNI program	Women who consumed 500 mg/d calcium tablets for more than 6 months during pregnancy had a 45% lower risk of developing hypertension compared to those who consumed less calcium (RR = 0.55, 95% CI = 0.33–0.93	CHW	No significant difference in the covergae of four or more ANC visits in PIH or non-PIH women
Saville, et al. (2018) [40]	Four-arm cluster RCT	I: PLA + food: 2997; PLA +cash: 3065; PLA only: 2448; C: 2426	Pregnant women	Nepal	I: Arm 1: PLA only Arm 2: PLA + food supplement Arm 3: PLA + cash transfer C: current gov’t program	Compared to the control arm, mean BW significantly higher in the PLA + food arm, by 78.0 g (95% CI 13.9, 142.0) and not in others; no significant difference in any other outcome (WAZ, LAZ, WLZ, HC, maternal BMI, MUAC, and IYCF)	Female community health volunteers + nutrition mobilizers (incentivized)	Enhanced participation in women’s groups increased institutional delivery rate in the intervention group
More, et al. (2018) [41]	Mixed-method, quasi-experimental, cross-sectional design	I: 3455 Children C: 2122 Children	Pregnant women and children under age 3	Mumbai, India	I: growth monitoring of 0–6 years children + home visits and counselling + CBMNT distribution + health camps + referrals + group meetings and events C: routine ICDS services	Prevalence of wasting decreased by 28% (18% to 13%) in intervention areas and by 5% (16.9% to 16%) in comparison areas; children in intervention areas significantly less likely to be malnourished (adjusted odds ratio, 0.81; confidence interval, 0.67 to 0.99)	Frontline health workers	High levels of coverage and lower levels of wasting, particularly severe wasting, in the program intervention areas.
Stevens, et al. (2018) [42]	Mixed-method, cross sectional survey and qualitative study design	Pregnant Women: baseline = 371, end line = 307; Local Health Workers: baseline = 100, end line = 79	Pregnant women and local health workers	Thailand-Myanmar	I: community-based participatory action plan (workshops for health workers + posters in centers + pamphlets distribution + presentations + small group discussions)	No significant improvement in preconception folic acid uptake; however, substantial increase in local healthcare workers’ knowledge	Local health workers (medics, midwives, nurses + ultrasound workers + basic healthcare workers)	No significant uptake of preconception folic acid
Stevens, et al. (2018) [43]	Village-matched cluster RCT (3rd phase of a multiphase RCT)	I: 58 C: 29	Undernourished pregnant women with MUAC of ≤22.1 cm	Northern Bangladesh (rural)	I: nutrition screening + nutrition education + ANC/PNC services + supplements C: nutrition screening + nutrition education + ANC/PNC services	MUAC significantly larger in infants of mothers in the intervention group compared to control group at 6 months (*p* < 0.05). Mean BW in babies of supplemented mothers (mean: 2.91 kg; SD: 0.19) higher than in babies of mothers in control group (mean: 2.72 kg; SD: 0.13); proportion of LBW babies in the intervention group was much lower (event rate = 0.04) than in the control group (event rate = 0.16). However, none of these differences are statistically significant (*p* > 0.05), most likely due to small sample size. The intervention reduced the risk of wasting at 6 months by 63.38% (RRR = 0.6338) and of low birth weight by 88.58% (RRR = 0.8858), with NNT of 2.22 and 6.32, respectively.	Female community nutrition volunteers and one male and one female supervisor	Higher registration of women within the first trimester
Svefors, et al. (2018) [44]	Factorial randomized trial (Nov 2001 to Feb 2009)	E60Fe: 738, EMMS: 740, E30Fe: 739, U60Fe: 741, UMMS: 741, U30Fe: 741	Pregnant women	Bangladesh (rural)	I: MINIMat trial of food supplementation ^⁑^ (E30Fe, E60Fe, EMMS, U30Fe, U60Fe, UMMS)	By incremental U60Fe to EMMS, one disability adjusted life years, averted at a cost of USD 24	Community volunteers	Not reported
Wendt, et al. (2018) [45]	Cross-sectional, observational, mixed-method (Nov 2011 to July 2012)	IDI: 59 (health workers at state, district, block, health sub-centre, and village levels) ANM survey: 340	Health workers at state, district, block, health sub-centre, and village levels; ANM	Bihar, India	-	44% of ANM were out of IFA stock. Stock levels and supply-chain practices varied greatly across districts. Specific bottlenecks impacting IFA were forecasting, procurement, storage, disposal, lack of personnel, and few training opportunities for key players in the supply chain	ASHA, AWW, ANM	Not reported
Pavithra, et al. (2019) [46]	Community-based intervention study (December 2012 to October 2014)	I: 64 children C: 64 children	57 mothers and 60 mothers of 64 moderate and severely malnourished children aged 13–60 months in the intervention group and control group, respectively	Puducherry, India (rural)	I: one-to-one communication with mothers concerning their child’s nutritional status and growth monitoring; education and child feeding practices; reinforcement of contents of health education	Awareness level in all domains increased significantly in the intervention group; 81% (52) of malnourished children turned out normal, whereas in the control group, 64% (41) of became normal; statistically significant difference between the mean changes in protein intake among boys (15.34 g to 19.91 g in the intervention group against 13.6 g to 16.24 g in the control group) and girls (15.09 g to 19.57 g in the intervention group against 13.36 g to 16.51 g in the control group), as well as calorie intake among girls (993.86 kcal to 1116.55 kcal in the intervention group against 992.65 kcal to 1078.75 kcal in the control group) between the two groups	Unclear	Not reported
Dhaded, et al. (2020) [47]	Secondary analysis; the parent study was an individually randomized, non-masked, multi-site randomized controlled efficacy trial	I: LBM-PC: 1281; LBM-FT: 1277 C: 1280	Mothers and their children (newborns)	India and Pakistan (rural)	I: Arm 1: received LBM at least 3-months prior to conception; Arm 2: received LBM near the end of the 1st trimester; additional protein-energy supplement was given to women whose BMI was <20 kg/m^2^ for both Arm 1 & 2 till delivery	LBM-PC associated with a decrease of 44% in stunting, 24% in wasting, and 26% SGA when compared to the control group; the difference between LBM-FT and control group was marginal	Home visitor research assistants	Increased compliance with supplements in the intervention arms (more in the first arm than the second)

Abbreviations: ANC: antenatal care; ANM: auxiliar nurse midwife; ASHA: accredited social health activist; AWW: Anganwadi workers; BMI: body mass index; BW: birth weight; 95% CI: 95% confidence interval; C: control group; CBMNT: community-based medical nutrition therapy; CCT: conditional cash transfer; CHW: community health worker; DBT: direct beneficiary transfer; E30Fe: early invitation with 30 mg iron and 400 μg of folic acid; E60Fe: early invitation with 60 mg iron and 400 μg of folic acid; EMMS: early invitation with multiple micronutrients; U30Fe: Usual invitation with 30 mg iron and 400 μg of folic acid; U60Fe: usual invitation with 60 mg iron and 400 μg of folic acid; UMMS: usual invitation with multiple micronutrients; EBF: exclusive breastfeeding; FGD: focus-group discussions; HC: head circumference; I: intervention group; IFA: iron folic acid; IYCF: infant and young-child feeding; ICDS: Integrated Child Development Service Scheme; IDI: in-depth interview; LAZ: length-for-age Z scores; LNS: lipid-based nutrient supplements; LBM-PC: lipid-based micronutrients at least 3 months prior to conception; LBM-FT: lipid-based at the end of the first trimester; LBW: low birth weight; MNCH: maternal and newborn child health; MDD; minimum dietary diversity; MNI: maternal nutrition initiative; MNP: micronutrient powder. MUAC: mid-upper-arm circumference; NNT: number needed to treat; NGO: non-governmental organization; OR: odds ratio; PLA: participatory learning and action; PW: pregnant women; PIH: pregnancy-induced hypertension; PNC: postnatal care; RDEARs: relative dietary energy adequacy ratios; pp: percentage points; RDW: recently delivered women; RCT: randomized controlled trial; RR: relative risk; RRR: relative-risk reduction; SGA: small for gestational age; SD: standard deviation; WAZ: weight-for-age Z scores; WLZ: weight-for-length Z scores. ^¶^ Nutrition-focused MNCH included greater specificity of interpersonal counseling, provided free supplements, conducted weight-gain monitoring during pregnancy, engaged fathers more explicitly, and included community mobilization activities. ^α^ IFA-MNP: women who received iron and folic acid during pregnancy and the first 3 months postpartum, and children received micronutrient powder from 6–24 months of age; IFA-LNS: women received iron and folic acid during pregnancy and the first 3-months postpartum and children received lipid-based nutrient supplements from 6–24 months of age; LNS-LNS: women and children received lipid-based nutrient supplements; IFA-control: women who received iron folic acid supplements during pregnancy and the first 3 months postpartum, and children did not receive any supplements. * A PLA cycle had four phases: identify problem, plan strategies, act together, and evaluate impact; In the first phase, groups used participatory methods, such as picture cards, games, and stories, to discuss nutrition problems and local barriers to achieving good health during pregnancy. In the second phase, groups prioritized and voted on the issues they wanted to focus on, designed strategies to address these problems, and engaged the wider local community for support and feedback. In the third phase, the groups implemented these strategies while continually discussing new topics related to pregnancy and infant health. Finally, in the fourth phase, the groups reviewed what went well and discussed what to do next after the implementing organization withdrew from the community. ^⁑^ MINIMat trial consisted of six intervention groups, namely E30Fe, E60Fe, EMMS, U30Fe, and U60Fe, UMMS; early invitation (E, at about 9 weeks of pregnancy), or usual timing (U, at about 20 weeks of pregnancy) with 600 kcal six days per week. Further, there were three separate micronutrient groups given from 14 weeks of gestation: 60 mg iron and 400 μg folic acid (routine); multiple micronutrients (MMS) with 15 micronutrients, including 30 mg iron and 400 μg folic acid; or 30 mg iron and 400 μg folic acid to control for the lower amount of iron in the MMS supplement.

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
