# Peer review of "Scoping Review of Intervention Strategies for Improving Coverage and Uptake of Maternal Nutrition Services in Southeast Asia"

_ijerph, 2021, doi:10.3390/ijerph182413292_

Round 1

Reviewer 1 Report

This paper provides a summary of a selected set of published papers dealing with improving the health , of pregnant women in Southeast Asia.  Its goal is to distill from those papers some conclusions about what types of implementation of programs put in place improves “coverage and takeup” of program services.  It provides brief summaries of outcomes, characteristics, and types of workers used in each of the 22 studies.  Then there is a non-rigorous review of studies categorized by the type of program (program goals or physical interventions (nutrition, deworming treatments, etc.).

The intended novelty in this summary is that it is focused on implementation rather than on the effectiveness of a program if it were fully implemented.  I expected to learn what program characteristics, especially the type of worker used, led to greater or lesser takeup or size of population that actually used the intervention.  My main criticism is that any such evidence is difficult to identify in the informal discussion of how programs were structured.  I did benefit from reading the summaries of the included studies but was not helped to decide which interventions had higher takeup than average, or even just high takeup by some standard, as a function of how the program was implemented.

Here is an example of what I thought I might learn.  Did programs implemented by the use of community health workers (CHWs) have higher takeup and coverage than those that used other kinds of workers?  The verbal summary says “almost all programs used CHWs’.  If this is true, there is no way to judge effectiveness relative to some other alternative type of staffing (community leaders, government workers, etc.) In fact it seems that some of the programs did not use CHWs, but with a small sample of studies to begin with it is hard to draw any conclusions about the difference CHWs make.  This challenge is compounded by the fact that most studies apparently did not report takeup but reported health outcomes (which will depend on the nature of the intervention as well as on its takeup). 

Some rewriting that is linked specifically to takeup and its correlates across studies might solve this problem but that would require major revision.  Perhaps it would also benefit if the authors used some concepts from the emerging scholarly field of implementation science.

There is also the limitation that interventions which failed to change outcomes are going to be less likely to be published than those which succeeded; there will be a problem of “publication bias”.  More successful outcomes associated with CHWs might just happen if more interventions used CHWs than any other kind of staffing.

Apparently there were almost no studies which used novel methods to improve takeup such as radio and television announcements, billboards, messaging in association with other governmental activities (such as political meetings, when mail or government funds are picked up, etc.) 

The conclusions from the paper seem eminently sensible though here again they do not seem to be linked to takeup or other metrics for implementation but to whether on not the program  reported statistically significant results relative to a comparator.  There was no information I saw on the relationship of implementation features to the quantitative size of takeup (eg, what adds 10% more to the population that actually obtained and used the intervention).  

Author Response

This paper provides a summary of a selected set of published papers dealing with improving the health, of pregnant women in Southeast Asia.  Its goal is to distill from those papers some conclusions about what types of implementation of programs put in place improves “coverage and takeup” of program services.  It provides brief summaries of outcomes, characteristics, and types of workers used in each of the 22 studies.  Then there is a non-rigorous review of studies categorized by the type of program (program goals or physical interventions (nutrition, deworming treatments, etc.).

The intended novelty in this summary is that it is focused on implementation rather than on the effectiveness of a program if it were fully implemented.  I expected to learn what program characteristics, especially the type of worker used, led to greater or lesser takeup or size of population that actually used the intervention.  My main criticism is that any such evidence is difficult to identify in the informal discussion of how programs were structured.  I did benefit from reading the summaries of the included studies but was not helped to decide which interventions had higher takeup than average, or even just high takeup by some standard, as a function of how the program was implemented.

Here is an example of what I thought I might learn.  Did programs implemented by the use of community health workers (CHWs) have higher takeup and coverage than those that used other kinds of workers?  The verbal summary says “almost all programs used CHWs’.  If this is true, there is no way to judge effectiveness relative to some other alternative type of staffing (community leaders, government workers, etc.) In fact it seems that some of the programs did not use CHWs, but with a small sample of studies to begin with it is hard to draw any conclusions about the difference CHWs make.  This challenge is compounded by the fact that most studies apparently did not report takeup but reported health outcomes (which will depend on the nature of the intervention as well as on its takeup). 

Some rewriting that is linked specifically to takeup and its correlates across studies might solve this problem but that would require major revision.  Perhaps it would also benefit if the authors used some concepts from the emerging scholarly field of implementation science.

There is also the limitation that interventions which failed to change outcomes are going to be less likely to be published than those which succeeded; there will be a problem of “publication bias”.  More successful outcomes associated with CHWs might just happen if more interventions used CHWs than any other kind of staffing.

Apparently there were almost no studies which used novel methods to improve takeup such as radio and television announcements, billboards, messaging in association with other governmental activities (such as political meetings, when mail or government funds are picked up, etc.) 

The conclusions from the paper seem eminently sensible though here again they do not seem to be linked to takeup or other metrics for implementation but to whether on not the program  reported statistically significant results relative to a comparator.  There was no information I saw on the relationship of implementation features to the quantitative size of takeup (eg, what adds 10% more to the population that actually obtained and used the intervention).  

Reply: Thanks reviewer for the comment. We have tried to address it. We have added an additional column in the tables to specify the coverage/uptake increased. However, I want to highlight that we did not intend to compare studies or approaches. As stated in the objective, we wanted to report different strategies and their effect on outcomes and not compare one strategy with the other. The objective of the review was to explore different strategies to improve maternal and child health outcomes. We have covered the delivery mechanism that addresses the ‘how part’ of the effect of the intervention on the uptake/coverage. Furthermore, the detailed result description added this component of the ‘how part’. Also, we added a para for the same in the discussion section. I have also added the respective part in the conclusion section. It is difficult to compare the effectiveness of the different strategies based on the quantitative size (we have not performed meta-analysis). On meta-analysis, we could have compared the effect of one intervention with the other based on the quantitative size. But the objective of our study was not to do meta-analysis.

We added:

“Studies lack data on the direct effect of health workers on the uptake of ser-vices/compliance (as revealed through association/regression/attribution) [12,18,19,23,24,26,29,32-35,37,38,42,43]. These studies compared the effect of the intervention (as a whole) on the outcomes without delineating the isolated effects of health workers/delivery mechanisms. However, some studies reported the impact of CHW on health outcomes. In MANOSHI program, CHW/trained providers/medically trained providers who paid monthly antenatal and postnatal visits reported greater effect of CHW on improving the uptake of 4 or more ANC visits and quality ANC visits than the routine services [28]. Frontline workers like ASHA/AWW who were of the same caste as the head of the household and were living in the same catchment area of pregnant women had higher odds of providing immunization services to under-5 and pregnan-cy-related counselling, respectively [27]. The CHW trained on additional curriculum of newborn care and with improved counselling skills enhanced maternal and newborn care practices significantly compared to those who did not receive additional training and skill development [25]. However, CHW in addition to existing AWW were not found to effect child health outcomes significantly [36].”

Reviewer 2 Report

See attached Word document

Author Response

Comments #1: Overall comments and results: Thank you for pulling together some interesting research in an effort to summarise evidence on a very important topic around improving coverage and uptake of maternal nutrition services in SE Asia. While I agree that the research question is really important, many of the interventions presented are purely maternal nutrition interventions (e.g. MMN vs. IFA supplementation trials) which do not incorporate interventions to improve coverage or uptake of supplantation. Either these interventions do not align with the objectives of the paper, or the way they are presented does not cover characteristics or results that meet the objectives. Some studies do present interventions targeting intervention uptake via strategies such as social marketing campaigns/education sessions etc., but in these cases, more detail is needed on how uptake/coverage is improved by these interventions rather than purely reporting on the overall effects on delivery/maternal outcomes. The included studies would need to be reconsidered or appropriately presented (e.g. are there intervention aspects targeting uptake/compliance etc that have not been reported) and the results should also be revised to present and discuss findings which specifically speak to the objectives and draw evidence-based conclusions. For example, you in some cases you state that BEP supplementation improved birth outcomes, but it is not clear whether/how coverage/compliance etc is considered here. Similarly, you state that community health workers, female health workers, research assistants, and community volunteers are crucial for delivering nutrition supplements, but there is a lack of data to support how interventions delivered via these health workers improve compliance/coverage etc in comparison to interventions that do not have these components. I think a major revision of the results is needed here, with only findings which speak to the objectives being presented and discussed, following which the conclusions should be revise accordingly.

Reply: Thanks reviewer for the comment. We have tried to address it. We have added an additional column in the tables to specify the coverage/uptake increased. However, I want to highlight that we did not intend to compare studies or approaches. As stated in the objective, we wanted to report different strategies and their effect on outcomes and not compare one strategy with the other. We have covered the delivery mechanism that addresses the ‘how part’ of the effect of the intervention on the uptake/coverage. Furthermore, the detailed result description added this component of the ‘how part’. Also, we added a para for the same in the discussion section.

“Studies lack data on the direct effect of health workers on the uptake of ser-vices/compliance (as revealed through association/regression/attribution) [12,18,19,23,24,26,29,32-35,37,38,42,43]. These studies compared the effect of the intervention (as a whole) on the outcomes without delineating the isolated effects of health workers/delivery mechanisms. However, some studies reported the impact of CHW on health outcomes. In MANOSHI program, CHW/trained providers/medically trained providers who paid monthly antenatal and postnatal visits reported greater effect of CHW on improving the uptake of 4 or more ANC visits and quality ANC visits than the routine services [28]. Frontline workers like ASHA/AWW who were of the same caste as the head of the household and were living in the same catchment area of pregnant women had higher odds of providing immunization services to under-5 and pregnan-cy-related counselling, respectively [27]. The CHW trained on additional curriculum of newborn care and with improved counselling skills enhanced maternal and newborn care practices significantly compared to those who did not receive additional training and skill development [25]. However, CHW in addition to existing AWW were not found to effect child health outcomes significantly [36].”

Comment #2: Some additional specific comments on language, as well as the abstract, introduction and methods are provided below: Language I would suggest that the authors review the manuscript for English language and grammar as the writing often makes it difficult to understand what is meant by the authors. For example in lines 58- 60 you state: Though studies on the efficacy and effectiveness of these intervention strategies have been conducted in the past; however, to the best of our knowledge, a scoping review of the effectiveness of delivery mechanism(s) of such evidence-based maternal nutrition interventions within the South East Asia region is the first of its kind. This phrasing is unclear and should be revised to make it more understandable to the reader.

Reply: Thanks reviewer. We have edited it. “Studies on the efficacy and effectiveness of these intervention strategies have been conducted in the past. However, to the best of our knowledge, we lacked a review of the effectiveness of delivery mechanism(s) of such evidence-based maternal nutrition interventions within the South East Asia region.”

Comment #3: Abstract - Similar to the above, the abstract should be revised to make the meaning clearer to the reader. For example, ‘various kinds of morbidities and even mortality’ in line 10 is very ambiguous and could be made clearer.

Reply: Yes, we have now changed it to “Maternal undernutrition leads to protein energy malnutrition, micronutrient deficiencies, or anemia during pregnancy or after birth.”

Comment #4: Similarly in line 21-23 you state that various interventions are effective, but what do you mean by this/how is this assessed?

Reply: Yes, we have now replaced it and made it more specific. “We need to incorporate diverse strategies, including different health education approaches, supplementation, strengthening community participation, and health system’s response for impactful maternal nutrition programs.”

Comment #5: Can you provide any additional detail, e.g. effective in improving what? I think this speaks to my above point on the results, so perhaps more clarity can be provided after the results have been revised.

Reply: Yes, addressed it already.

Comment #6: - Line 20: a minor point, but I would suggest revising this to state: “two from BOTH Pakistan and Thailand”

Reply: Thanks reviewer, yes, we have changed it.

Comment #7: Introduction - Line 29-30: This is very broad. Can you be more specific in terms of nutrition-related exposures, as well as whether you are referring to maternal outcomes, infant outcomes or both?

Reply: Thanks reviewer. I have now specified it and made it clear. “A shortfall in nutrition during these periods can lead to protein energy malnutrition, malnutrition deficiencies, poor weight gain or anemia.”

Comment #8: Line 36: I would avoid using chronic energy deficiency as a reported indicator - are you referring to underweight here? Or is this around intake?

Reply:  Yes, I meant to use underweight as this was used in the paper from where I took the reference.

Comment #9: Line 37: Why are the findings on India, which follow on from the previous, a new paragraph?

Reply: Now, I have merged India’s findings with other south east Asian countries in one para and not starting from a new para.

Comment #10: Line 42: Can you expand on what you mean by a ‘multi-pronged approach’

Reply:  Actually, by multi-pronged approach, I meant to specify the approach with multiple aspects/elements. Like, to address maternal undernutrition, we need to ensure food security, good antenatal care and counselling, iron folic acid supplementation and supplementation of other micronutrients, improve poverty levels, and hygiene, and improve knowledge about good diet and foods, etc.

Comment #11: Line 51-54: In what populations is BEP supplementation recommended? This makes it sound like global blanket supplementation is recommended, but this is not the case.

Reply: Thanks reviewer for the comment. Yes, BEP recommended for undernourished pregnant women. We have specified that now.

Comment #12: Line 61-63: I would recommend moving this up to where you discuss implications for outcomes above

Reply: Based on your suggestion, I have moved this up.

Methods

Comment #13: Line 73: In the research question you refer to looking at what are the strategies. Were you not looking at the effectiveness if the strategies too? This should be clear from the research question

Reply: Thanks reviewer for the comment. Yes, we have now included the word ‘effectiveness’.

Comment #14: Supplementary material refers to the search terms as a figure, but this is a box? Have you provided a completed PRISMA checklist with the scoping review?

Reply: Yes, we have now changed it to Supplementary box. No, actually at the time of working on the paper, we did not get the checklist for the scoping review, though we had the checklist for the systematic review. So, we did not provide the PRISMA checklist for the scoping review.

Comment #15: Line 81-83 is very repetitive. Suggest revising this - It would be useful to list inclusion/exclusion criteria clearly, as these aren't completely clear from the narrative

Reply: Thanks reviewer. Yes, we have now edited it and shortened this line.

Comment #16: Line 91: Do you mean outcome of the study rather than this review? Please clarify this sentence

Reply: Yes, we have now changed the word ‘review’ to ‘study’.

Comment #17: Line 115: Not very clear what you mean here in relation to strategy vs. intervention and how results were presented. Results/conclusion – see overall comments above

Reply: Thanks reviewer for highlighting this. Yes, we have now deleted the line to avoid confusion.

Reviewer 3 Report

Abstract:

After summarizing results in the abstract section, write a strong key message or conclusion at the end of the abstract.  The authors mentioned that "effective, context-specific intervention strategies and delivery mechanisms need to be incorporated for impactful maternal nutrition program implementation" which does not provide a key message from this study. This sentence is very generic. 

Line#31 and 32 mention percentage in parenthesis along with the frequency, for example, 450 million (%) women are estimated to have short 
stature, 240 million (%) are underweight........... for better understanding the context of the problem.

Line#35 40% women are anemic is difficult to understand either 40% of women from Pakistan or Indonesia are anemic? Please clarify. Moreover, these findings are the part of the below study instead of reference 01 that you mentioned.

Atmarita: Nutrition problems in Indonesia; in: An Integrated International Seminar and Workshop on Lifestyle-Related Diseases 2005; Gajah Mada University, 19–20 March 2005. Directorate of Community Nutrition, Ministry of Health, 2005

 Moreover, I would suggest making your case first for southeast Asian countries for maternal nutrition services. Then do focus on key countries that have maternal malnutrition or nutritional deficiency burden.

Before starting from L#42, summarize what types of interventions have been rolled out in the last decades/past and what was their impact? What were major gaps in those interventions and then identify the gaps and justify the need for this scoping review.  

Material and Methods

Maybe authors can write the research question in the past tense. For example, what were the strategies..... 

2.6. Collating, summarizing, and reporting the results

Line#116-117. Are you referring to some table that is part of this paper? if yes, then refer to that table over here too.

Data extraction process:

Authors can align these themes with research questions. (a) study characteristics, (b) methodological characteristics, (c) intervention strategies, and (d) targeted outcome of this review. 

Maybe the authors can add another supplementary research question.

Results

After the title and abstracts were reviewed, 231 articles were excluded.

Explain/summarize a brief reason for the exclusion of these studies. Moreover, in each step of exclusion of studies, provide a sentence summary for reduction of studies. 

There was no disagreement between the two reviewers. For what? For ensuring the quality of the papers?

I would suggest merging tables 1a and 1b into one table. And the author may add table 02 in which authors can only mention study intervention. In such a way the tables will be small and brief. The author may not follow this instruction of the reviewer. 

Strengthen and limitation: this section can be further strengthened to enlist strengthens and limitations and how both influenced this study.

Authors can summarize which types of interventional strategies were adopted in South East Asian countries.

And moreover, these findings can also be summarized in the conclusions section. 

Author Response

Comment #1: After summarizing results in the abstract section, write a strong key message or conclusion at the end of the abstract.  The authors mentioned that "effective, context-specific intervention strategies and delivery mechanisms need to be incorporated for impactful maternal nutrition program implementation" which does not provide a key message from this study. This sentence is very generic.

Reply:  Thanks for highlighting this. We have now replaced the sentence in the abstract. “We need to incorporate diverse strategies, including different health education approaches, supplementation, strengthening community participation, and health system’s response for impactful maternal nutrition programs.”

Comment #2: Line#31 and 32 mention percentage in parenthesis along with the frequency, for example, 450 million (%) women are estimated to have short stature, 240 million (%) are underweight........... for better understanding the context of the problem.

Reply: We took these figures from a paper. However, the paper did not specify the percentages. We cross-checked it with others, but could not find the percentages. However, we did not prefer self-calculation of the percentages.

Comment #3: Line#35 40% women are anemic is difficult to understand either 40% of women from Pakistan or Indonesia are anemic? Please clarify. Moreover, these findings are the part of the below study instead of reference 01 that you mentioned.

Atmarita: Nutrition problems in Indonesia; in: An Integrated International Seminar and Workshop on Lifestyle-Related Diseases 2005; Gajah Mada University, 19–20 March 2005. Directorate of Community Nutrition, Ministry of Health, 2005

Reply: Thanks reviewer for the comment. To clarify, 40% of women are anemic in Indonesia. I have added the reference as suggested.

Comment #4: Moreover, I would suggest making your case first for southeast Asian countries for maternal nutrition services. Then do focus on key countries that have maternal malnutrition or nutritional deficiency burden.

Reply: Thanks for highlighting this. Yes, as per your suggestion, we have added the respective part in Para 1 of the introduction session and then moved to specific countries from the next para.

Comment #5: Before starting from L#42, summarize what types of interventions have been rolled out in the last decades/past and what was their impact? What were major gaps in those interventions and then identify the gaps and justify the need for this scoping review.  

Reply: Thanks reviewer for the comments. I have now added a para on this. I have specified some of the interventions used in the past, and the gaps identified.

“A review in the past highlighted that interventions emphasizing counselling or delivering information to women or their family members improve nutrition outcomes like the consumption of iron-folic acid and calcium tablets by pregnant women. Using counselling material based on locally relevant information and engaging influencers like mothers-in-law and husbands are effective approaches to improve maternal nutrition outcomes. Furthermore, home visits and using community forums for greater participation enhance access to services by pregnant women. However, digital interventions in silos are not proven as an effective approach. The studies included in the review were limited and focused on either IFA or calcium supplementation or access to information on intake of diet. Also, most of the available evidence reported in the review was from India and limited from other countries like Pakistan, Indonesia, or Bangladesh.”

Material and Methods

Comment #6: Maybe authors can write the research question in the past tense. For example, what were the strategies..... 

Reply: Yes. I have now changed it into the past tense.

Comment #7: 2.6. Collating, summarizing, and reporting the results

Line#116-117. Are you referring to some table that is part of this paper? if yes, then refer to that table over here too.

Reply: Thanks for highlighting this. Actually, I meant to tell the readers that we drew a table to display the included studies. But I removed the sentence to avoid confusion. There is a mention about the supplementary table later in the text.

Data extraction process:

Comment #8: Authors can align these themes with research questions. (a) study characteristics, (b) methodological characteristics, (c) intervention strategies, and (d) targeted outcome of this review. 

Maybe the authors can add another supplementary research question.

Reply: Thanks reviewer for the comments. I have now added a supplementary research question.

‘What was the effectiveness of the different strategies for delivering facility- or com-munity-based maternal nutrition intervention/programs in countries of South East Asia?’ Also, what were the outcomes, methodological approaches, and characteristics of the intervention/programs.

Results

Comment #9: After the title and abstracts were reviewed, 231 articles were excluded.

Explain/summarize a brief reason for the exclusion of these studies. Moreover, in each step of exclusion of studies, provide a sentence summary for reduction of studies. 

Reply: Thanks reviewer. I have specified the reasons of exclusion of studies after reading the full texts. For other steps of exclusion studies, the major issue was that the studies did not meet the inclusion criteria.

Comment #10: There was no disagreement between the two reviewers. For what? For ensuring the quality of the papers?

Reply: Thanks reviewer for highlighting this. Yes, I have now added that “There was no disagreement between the two reviewers for the inclusion of paper based on the quality checks.”

Comment #11: I would suggest merging tables 1a and 1b into one table. And the author may add table 02 in which authors can only mention study intervention. In such a way the tables will be small and brief. The author may not follow this instruction of the reviewer. 

Reply: Actually, merging two tables would make the entire table lengthy and difficult for the reader to understand.

Comment #12: Strengthen and limitation: this section can be further strengthened to enlist strengthens and limitations and how both influenced this study.

Reply: Thanks Reviewer for the comments. We have now added limitations and strengths.

“Also, meta-analysis was not done in the present paper due to varied outcome indicators and non-uniformity in the presentation of results across studies. However, the studies included in the review present a diverse range of strategies to improve maternal nutrition from many countries, including India, Pakistan, Indonesia, etc., compared to the previous reviews. Furthermore, many aspects of maternal nutrition programs were discussed, including strategies, effectiveness to improve maternal and child health outcomes, and delivery mechanisms. This makes the paper relevant for policy-makers, public health specialists, and academics.”

Comment #13: Authors can summarize which types of interventional strategies were adopted in South East Asian countries.

And moreover, these findings can also be summarized in the conclusions section. 

Reply: We have summarized most of the intervention strategies that worked well in the South-East Asian countries. Further, I have added some more points in the conclusion section.

Reviewer 4 Report

This scoping qualitative review examines intervention strategies successfully improving coverage and uptake of maternal nutrition services and programs in developing countries.

The search window of this scoping review was restricted to about 10 years, from 7 November 2010 to 17 February 2021. Eventually  22 studies were included, as follows:  12  studies from India, 11 from Bangladesh, 3 from Nepal, 2 from  Thailand and one from Indonesia, were included for narrative synthesis. Then manuscript is interesting well written and deserves publication. Effective strategies improving coverage and uptake of community services included use of  community health workers, domiciliary visits, directly observed nutrition supplementation; community mobilization; local fairs.

The study is interesting and the manuscript well written.

Please find a few minor points to be addressed.

Lines 37-38: it is worth mentioning that in India an overall improvement in the prevalence of both BMI < 18.5 and anaemia among adolescents nullipara was recently reported in a study using DHS. Please cite this recent publication on this matter [PMID: 33100257]. The same article should be cited to back the section “Effect of home visits by CHW on compliance to nutrition interventions” (Lines 413-38), discussing  the role of community services, which implies investments in universal health care are and improved access to primary care services to monitor the nutritional status of women [PMID: 33100257]

Lines 61-63: when discussing the impact of maternal malnutrition on low birthweight, it is worth citing this recent global health study on low birthweight estimates: [PMID: 31103470].

Lines 349-350: “outcomes aren’t adequate, but a few studies outcomes aren’t adequate, but a 349 few studies”… remove “but”

Lines 410-11: The present review qualitatively narrates various intervention strategies adopted to deliver interventions”, please rephrase to avoid repetition of the word “intervention”

Paragraphs 4.1, 4.2, 4.3 and 4.4 at times reiterate similar concepts. I wonder whether the discussion could be contracted, since the paper is rather long to follow.

Author Response

This scoping qualitative review examines intervention strategies successfully improving coverage and uptake of maternal nutrition services and programs in developing countries.

The search window of this scoping review was restricted to about 10 years, from 7 November 2010 to 17 February 2021. Eventually  22 studies were included, as follows:  12  studies from India, 11 from Bangladesh, 3 from Nepal, 2 from  Thailand and one from Indonesia, were included for narrative synthesis. Then manuscript is interesting well written and deserves publication. Effective strategies improving coverage and uptake of community services included use of  community health workers, domiciliary visits, directly observed nutrition supplementation; community mobilization; local fairs.

The study is interesting and the manuscript well written.

Please find a few minor points to be addressed.

Comment #1: Lines 37-38: it is worth mentioning that in India an overall improvement in the prevalence of both BMI < 18.5 and anaemia among adolescents nullipara was recently reported in a study using DHS. Please cite this recent publication on this matter [PMID: 33100257]. The same article should be cited to back the section “Effect of home visits by CHW on compliance to nutrition interventions” (Lines 413-38), discussing  the role of community services, which implies investments in universal health care are and improved access to primary care services to monitor the nutritional status of women [PMID: 33100257]

Reply: Thanks reviewer for the comments. I have added this reference at two places as suggested.

Comment #2: Lines 61-63: when discussing the impact of maternal malnutrition on low birthweight, it is worth citing this recent global health study on low birthweight estimates: [PMID: 31103470].

Reply: Thanks reviewer for the comment. Yes, I have added the reference. It is a valuable reference to add.

Lines 349-350: “outcomes aren’t adequate, but a few studies outcomes aren’t adequate, but a 349 few studies”… remove “but”

Reply: Thanks reviewer for the comment. Yes, I have removed the word ‘but’ from the sentence.

Lines 410-11: The present review qualitatively narrates various intervention strategies adopted to deliver interventions”, please rephrase to avoid repetition of the word “intervention”

Reply: Yes, I have now replaced the word ‘intervention’ with ‘program’.

Paragraphs 4.1, 4.2, 4.3 and 4.4 at times reiterate similar concepts. I wonder whether the discussion could be contracted, since the paper is rather long to follow.

Reply: Actually, we tried to keep it short but the looking at the vast result section, we had to discuss bare minimum points.